# Laminin N-terminus α31 regulates corneal epithelial cell adhesion and migration through modifying the organization and proteolytic processing of laminin 332

Lee D. Troughton[1⊙], Valentina Iorio[2*], Liam Shaw[3], Conor J. Sugden[4],
Natasha D. Chavda[3], Peter Wilson[3], Djamilla Simoens[1], Andrea E. Conway[3],
Stefano Sala[5], Simon Kaja [1], Kazuhiro Yamamoto[3,6], Kevin J. Hamill [3*]

1 Department of Ophthalmology, Stritch School of Medicine, Loyola University Chicago, Maywood, Illinois, United States of America, 2 Institute of Integrative Biology, University of Liverpool, Liverpool, United Kingdom, 3 Institute of Life Course and Medical Sciences, University of Liverpool, Liverpool, United Kingdom, 4 Division of Cell Matrix Biology & Regenerative Medicine, School of Biological Sciences, The University of Manchester, Manchester, United Kingdom, 5 Department of Cell and Molecular Physiology, Stritch School of Medicine, Loyola University Chicago, Maywood, Illinois, United States of America, 6 Scleroprotein and Leather Institute, Faculty of Agriculture, Tokyo University of Agriculture and Technology, Tokyo, Japan

⊙ These authors contributed equally to this work.
* khamill@liverpool.ac.uk (KJH); v.iorio@liverpool.ac.uk (VI)

## Abstract

Laminin N-terminus α31 (LaNt α31) is a netrin-like protein generated by alternative splicing of the laminin α3 gene. While previously shown to regulate vascular permeability in vivo, its role in epithelial tissues remains less defined. Here, we demonstrate that LaNt α31 modulates epithelial cell behavior by altering laminin 332 (LM332) organization and hemidesmosome (HD) maturation. Adenoviral driven overexpression of LaNt α31 in corneal epithelial cells led to premature HD assembly, marked by enhanced recruitment of collagen XVII and BPAG1e to β4 integrin, and reduced cell migration. LaNt α31 expression reorganized LM332 from diffuse arcs into tight clusters and co-localized with LMβ3 during matrix deposition. Notably, phenotypes were rescued by precoated extracellular matrix, indicating a matrix-dependent mechanism. Furthermore, LaNt α31 increased matrix metalloproteinase (MMP) activity and LMα3 proteolytic processing, both essential for its effects, as MMP inhibition reversed LM clustering and HD maturation. These findings identify LaNt α31 as a regulator of epithelial homeostasis through modulation of LM332 architecture and cell–matrix adhesion.

## Introduction

The laminins (LMs) are a family of essential structural proteins incorporated into specialized regions of the ECM known as basement membranes (BMs). In intact

**Data availability statement:** All relevant data are within the paper and its Supporting Information files.

**Funding:** This work was supported by funding from the Biotechnology and Biological Sciences Research Council (K.H BB/L020513/1 and BB/P025773/1, and a BBSRC DTP PhD studentship), Northwest Cancer Research Fund (K.H), Fight For Sight (K.H New lecturers' award), British Skin Foundation (K.H PhD studentship), Foundation for the Prevention of Blindness (K.H), ECMage impact award (K.H), Versus Arthritis Career Development Fellow (K.Y 21447), Versus Arthritis Bridging Fellow (K.Y 23137). Additional support from the National Institute of Health (S.K grant R24/EY032440), the Dr. John P. and Therese E. Mulcahy Endowed Professorship in Ophthalmology (S.K), the Richard A. Perritt M.D. Charitable Foundation and the Illinois Society for the Prevention of Blindness are gratefully acknowledged (L.T). The funders played no role in the study design, data collection and analysis, decision to publish, or preparation of this manuscript. https://www.ukri.org/councils/bbsrc/ https://www.fightforsight.org.uk/ https://www.britishskinfoundation.org.uk/ https://nwcr.org/ https://versusarthritis.org/ https://www.nih.gov/ https://eyehealthillinois.org/.

**Competing interests:** The authors have declared that no competing interests exist.

**Abbreviations:** LM –laminin, LaNt – laminin N-terminus proteins, LN – laminin N-terminal domain, LE – laminin-type epidermal growth factor-like, LCC – laminin coiled-coil domain, MMP – matrix metalloproteinase, HD – hemidesmosome, FA – focal adhesion, ECM – extracellular matrix, TIRF – total internal reflection microscopy, IF – immunofluorescence microscopy, IB – immunoblotting, IP – immunoprecipitation, DMEM – Dulbecco's modified Eagle's medium, KSFM – keratinocyte serum free media, BPE – bovine pituitary extract.

tissues, BMs provide architectural support for epithelial and endothelial sheets, nerves, and muscle [1,2]. LMs also regulate dynamic processes such as cell migration during tissue remodelling and wound repair by acting as extracellular ligands for actin-associated cell-surface receptors. Transitioning between stable adhesion and migration-supporting states requires context-specific remodelling of the LM substrate. Understanding the mechanisms that govern these transitions could inform therapeutic strategies, particularly in conditions where wound healing is delayed or ineffective, such as recurrent corneal erosion syndrome [3,4].

Laminin N-terminus α31 (LaNt α31) is a netrin-like protein produced by alternative splicing of the *LAMA3* gene [5]. It is widely expressed across tissues [5–7], with high abundance in the limbal epithelium and is upregulated during corneal wound healing [6]. Elevated LaNt α31 expression has also been observed in breast cancer and associated nodal metastases [8]. In vivo, transgenic overexpression of LaNt α31 caused vascular leakage across multiple tissues; however, in the epidermis, LaNt α31 expression led to increased HD size, suggesting altered cell–matrix adhesion and pointing to context/tissue-specific functions for LaNt α31 [9]. In vitro, LaNt α31 knockdown in epidermal keratinocytes reduced keratinocyte adhesion and delayed wound closure [5], while overexpression decreased migration of primary corneal-limbal epithelial cells [6].

Structurally, LaNt α31 contains an LN domain, several LE repeats, and a unique C-terminal tail, but lacks the coiled-coil domain required for LM trimerization [5]. This suggests it functions as an independent LM short-arm fragment. LN domains mediate LM network assembly through trimeric interactions between α, β, and γ chains [2,10–13]. Netrin-4, which shares partial homology with LM β chains, disrupts LM networks and reduces matrix stiffness [14–16]. While LaNt α31 may similarly modulate LM function, its role is complicated in epithelial tissues where LM332, the predominant LM, lacks a full complement of LN domains and cannot self-polymerize [11,17]. Despite this inability to form independent networks, LM332 is critical for epithelial BM integrity and wound repair [18–20], functioning through HDs that anchor cells via α6β4 integrin and type XVII collagen, linked to keratin filaments by plectin and BPAG1e [21–24]. HD maturation involves recruitment of these components and proteolytic processing of the LMα3 chain [25,26].

In this study, we investigated how LaNt α31 influences epithelial cell adhesion and migration in epithelial cells. We show that LaNt α31 alters LM332 organization and proteolytic processing, promotes HD maturation, and reduces cell migration, in a MMP-dependent manner. These findings reveal a novel mechanism of LM regulation with implications for epithelial tissue homeostasis.

## Results

### High LaNt α31 expression restricts epithelial cell migration during matrix synthesis

We previously showed that elevated LaNt α31 expression increases cell spreading and reduces migration in primary limbal-derived corneal epithelial cells [6]. To explore the mechanism, we used the limbal-derived corneal epithelial cell line, hTCEpi, a

slightly more mature corneal epithelial line based on K12 expression (S1 Fig) [27], and an adenoviral construct to express LaNt α31 with a C-terminal eGFP tag (+LaNt α31, Fig 1B), comparing it to eGFP-only controls (+eGFP).

Consistent with earlier findings, +LaNt α31 hTCEpi cells had over twice the 2D spread area of controls on uncoated plastic (Fig 1C-D; non-transduced: 839±41 µm², +eGFP: 943±33 µm², +LaNt α31: 2250±110 µm², $p<0.05$), measured ~12 h post-plating. Aspect ratios were similar across groups (Fig 1E; ~0.65–0.67). +LaNt α31 cells also showed significantly reduced migration speed (Fig 1F-G; non-transduced: 1.15±0.26 µm/min, +eGFP: 1.05±0.11 µm/min, +LaNt α31: 0.46±0.17 µm/min, $p<0.05$), with no change in processivity (Fig 1H; ~0.38–0.41). Gap closure assays also confirmed slower sheet migration (S2A-B Fig; closure after 16 h: non-transduced: 95±8%, +eGFP: 78±39%, +LaNt α31: 41±16%, $p<0.05$).

Coating wells with LM332 or fibronectin partially rescued migration, LM511 fully restored it, while collagen I had no effect (Fig 1I-J; motility: hTCEpi: 1.12±0.09 µm/min, +LaNt α31: 0.75±0.06, +LaNt α31 on fibronectin: 1.02±0.12, on collagen I: 0.73±0.05, on LM332: 0.91±0.07, on LM511: 1.41±0.48).

### LaNt α31 dynamically localises to LM332 clusters

Following on from the observed impact of LaNt α31 on epithelial cell migration and matrix-dependent motility, we next examined its spatial and temporal relationship with LM332. To assess LaNt α31's interaction with LM332 in live cells, we used an mCherry-tagged LMβ3 adenovirus. TIRF microscopy showed partial co-localisation of LaNt α31-eGFP and LMβ3-mCherry signals (Fig 2A), which was also observed in ECM extracts (Fig 2B). Live-cell imaging over 12 h revealed that LaNt α31-eGFP rapidly formed and disassembled clusters at LMβ3-mCherry deposition sites (Fig 2C, S1 File). To test for complex formation, we immunoprecipitated LaNt α31-eGFP and detected co-precipitated endogenous LMβ3, suggesting some form of complex formation (Fig 2D).

### LaNt α31 induces LM332 clustering and accelerates hemidesmosome maturation

Building on the dynamic association between LaNt α31 and LM332, we next investigated how increased LaNt α31 expression influences LM organisation and HD maturation. As enlarged HDs were observed in LaNt α31 transgenic mice [9], we first examined HD maturation *in vitro*. Indirect immunofluorescence analyses revealed increased colocalisation of the archetypal HD proteins type XVII collagen and BPAG1e with β4 integrin in +LaNt α31 cells, compared to minimal overlap in controls (Fig 3A-B; Manders' coefficient: non-transduced: 0.07±0.12, +eGFP: 0.22±0.17, +LaNt α31: 0.47±0.19, $p<0.05$).

To test whether LaNt α31 alters LM332 ECM organisation, we analysed LMα3 distribution. Control cells showed broad arcs and swirls, while +LaNt α31 cells formed tight clusters (Fig 3C-D; $p<0.001$). Transcript analysis confirmed LM3A32 as the dominant isoform in hTCEpi cells (S3A Fig), and immunoblotting showed no change in overall abundance of other LM chains(S3B Fig). However, ECM extracts revealed a ~2-fold increase in the processed LMα3 band (~160 kDa) relative to the unprocessed form (~190 kDa) in +LaNt α31 cells (Fig 3E-F; non-transduced: 0.4±0.2, +eGFP: 0.7±0.2, +LaNt α31: 1.3±0.5, $p<0.05$), consistent with HD maturation via LG4/5 domain cleavage [25,26].

While HDs influence migration [28–30], focal adhesions (FAs) are major migration-related force-generating structures [31]. Paxillin, a core FA adaptor, typically localises to discrete foci in control cells, but in +LaNt α31 cells, it redistributed into linear arrays (S4A Fig). This shift occurred without changes in phosphorylation of paxillin (Y118) or FAK (Y397) (S4B-E Fig).

### LaNt α31-induced hemidesmosome maturation requires MMP activity

Since induced LaNt α31 expression increased the proteolytic processing of LMα3, we next investigated MMP activity. Gelatin zymography revealed bands at sizes consistent with MMP2 and MMP9, with suggested elevated activity in +LaNt α31 cells (Fig 4A). This increased activity was confirmed using a fluorogenic MMP substrate, showing significantly higher

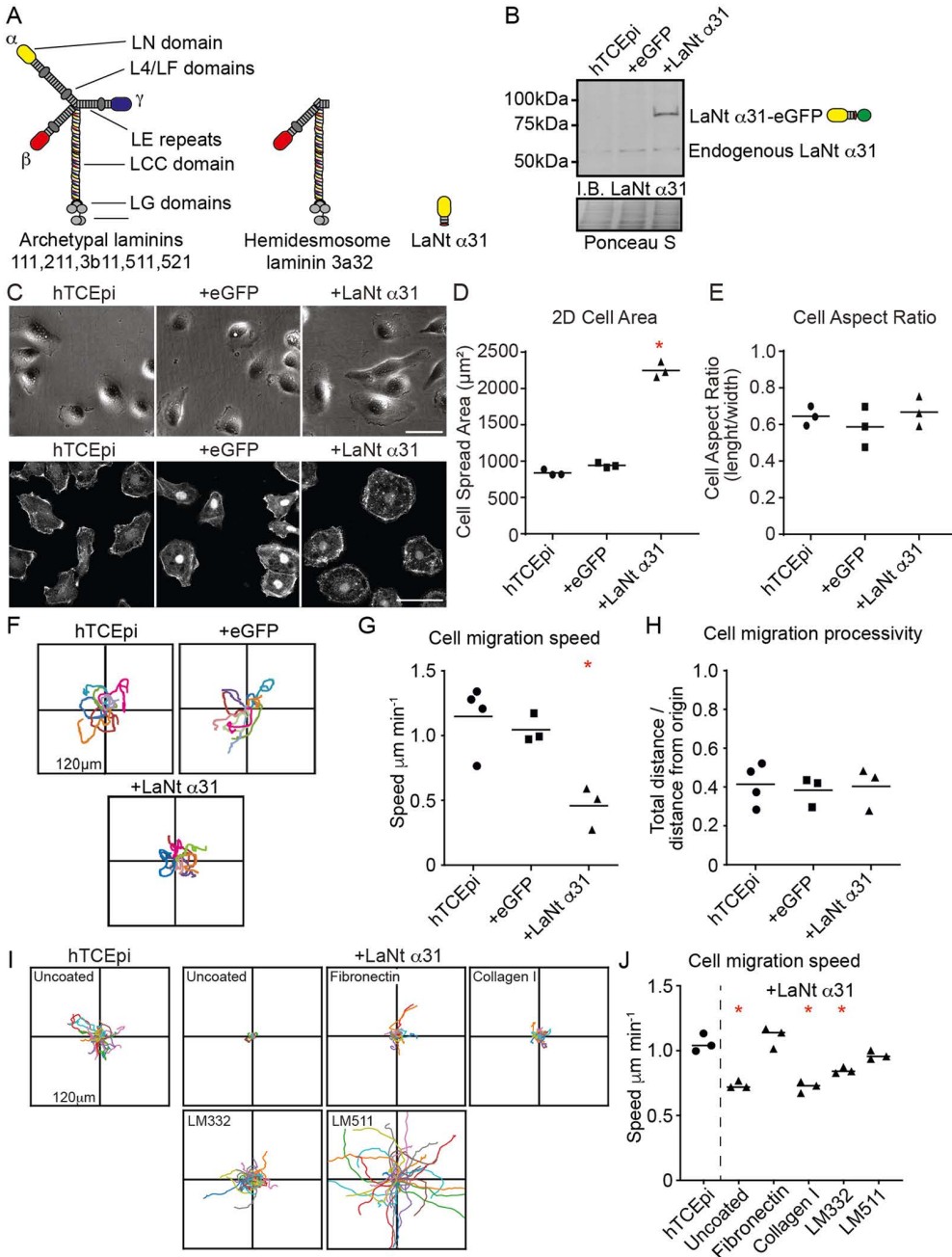

**Fig 1. Corneal epithelial cells induced to express LaNt α31display decreased cell migration, which can be rescued through provision of ECM.**
**(A)** Diagrammatic representation of archetypal laminins, hemidesmosome laminin 332, and LaNt α31 protein structure. LN – laminin N-terminal domain, LE – laminin-type epidermal growth factor-like repeats. LCC – laminin coiled coil domain, LG – laminin globular domain. **(B)** Immunoblot of total cell lysates generated from hTCEpi cells transduced with either eGFP (+eGFP) or LaNt α31-eGFP (+LaNt α31) then probed with antibodies against LaNt α31. **(C-H)** Non-transduced (hTCEpi), + eGFP, or +LaNt α31hTCEpi, were plated overnight at low density and either cell area and aspect area measured from live cells (C-E) or the migration paths of individual cells tracked over a 2 h period **(F-H)**. **(C)** Representative phase contrast images of live cells on tissue culture plastic (top), or cells on glass coverslips fixed and stained with phalloidin (bottom). Scale bars 50 μm. 2D cell area plotted from phase contrast images in **(D)**, and aspect ratio determined as length at longest point divided by width measured at 90" to that point **(E)**. Each point represents one independent biological experiment with 40-60 cells per cell type, with line at mean. **(F)** Vector diagrams showing representative paths of 10 individual cells with each colour representing a single cell. **(G)** Migration speed measured as total distance migrated over time. **(H)** Migration processivity; measured as total distance migrated divided by maximum distance from the origin. In each graph lines are at mean with each point representing an individual

experimental repeat consisting of the mean from 20-40 individual cells tracks. **(I-J)** Tissue culture plastic was pre-coated with 10 mg/ml of either fibronectin, col I, LM332, or LM511, and non-transduced or +LaNt α31 hTCEpi were seeded at low density, allowed to adhere for 4 h then the migration paths of individual cells tracked over a 2 h period. **(I)** Representative vector diagrams of migration paths of 10 randomly selected cells tracked over a 2 h period and (B) migration speed, with each point representing an individual experimental repeat consisting of the mean from >50 individual cells tracks. * denotes $p < 0.05$ compared with controls determined by one-way ANOVA followed by Bonferroni post hoc test.

fluorescence in +LaNt α31 media (Fig 4B; mean ± SD: non-transduced: 750 ± 70, + eGFP: 870 ± 98, + LaNt α31: 2980 ± 110, $p < 0.05$; $r^2$ values: 0.84, 0.78, 0.97 respectively).

To test whether the increase MMP activity was required to drive the LM332 clustering and HD maturation effects, + LaNt α31 cells were treated with either a broad-spectrum MMP inhibitor (CT-1746) or a non-MMP protease inhibitor mix (P1860). MMP inhibition reversed LMα3 processing and clustering (Fig 4C-D), with no difference in clustering between MMPi-treated +LaNt α31 and control cells (Fig 4E). Similarly, increased colocalisation of β4 integrin with type XVII collagen and BPAG1e was rescued by MMP inhibition but not by non-MMP inhibitors (Fig 4F-G). These results demonstrate that LaNt α31's effects on LM organisation and HD maturation are MMP-dependent.

## Discussion

This study revealed that elevated LaNt α31 expression in corneal epithelial cells lead to increased cell spreading and reduced migration, LM332 clustering and increased LMα3 processing, and enhanced HD assembly, all of which were reversed by MMP inhibition. Together, these data point to LaNt α31 as a hitherto unknown regulator of HD maturation and MMP-dependent matrix remodelling.

These findings position LaNt α31 as a modulator of epithelial plasticity, influencing the balance between motility and stable adhesion. The observed increase in cell spreading and HD maturation suggests that LaNt α31 promotes a more adhesive epithelial phenotype, potentially stabilizing cells in a stationary state. This is particularly relevant in the context of the corneal limbus, where epithelial stem and progenitor cells must remain anchored within a specialized niche [32]. The dynamic localization of LaNt α31 to LM332-rich regions and its ability to induce LM332 clustering suggest a role in organizing or stabilizing ECM microdomains, most likely via indirect interactions with LM components. The dependence on MMP activity for LMα3 processing and HD maturation adds a layer of complexity, linking LaNt α31 to protease-mediated ECM remodelling. LMα3 cleavage, particularly within the LG4/5 domains, is known to facilitate HD assembly by exposing integrin-binding sites [25,26]. Thus, LaNt α31 may act upstream in this pathway, either by recruiting or activating MMPs, or by altering LM332 conformation to enhance its susceptibility to proteolysis [33]. This mechanism echoes broader themes in epithelial biology, where ECM remodelling and protease activity are tightly coupled to changes in cell behaviour during wound healing, differentiation, and disease [34–37].

Interestingly, the matrix-specific rescue of migration, partial on LM332 and fibronectin, complete on LM511, highlights the context-dependent nature of LaNt α31's effects. LM511 is known to support epithelial migration and stem cell maintenance [38], suggesting that LaNt α31 may selectively modulate LM isoform function. This raises the possibility that LaNt α31 contributes to ECM specialization in different epithelial compartments, influencing cell fate and behaviour through selective matrix engagement. From a broader perspective, these findings suggest that LaNt α31 could serve as a molecular switch between migratory and adhesive epithelial states. This has potential implications for tissue repair, where transient suppression of migration and promotion of adhesion may be necessary for re-epithelialization and barrier restoration [39]. Conversely, dysregulation of LaNt α31 could contribute to pathological conditions such as fibrosis or epithelial cancers, where aberrant adhesion and matrix remodelling are hallmarks [40].

Future work should explore the molecular basis of LaNt α31's interaction with LM332 and MMPs, and whether its expression is dynamically regulated during epithelial remodelling in vivo. Understanding how LaNt α31 integrates with broader ECM and signalling networks may reveal new strategies for modulating epithelial behaviour in regenerative medicine and disease.

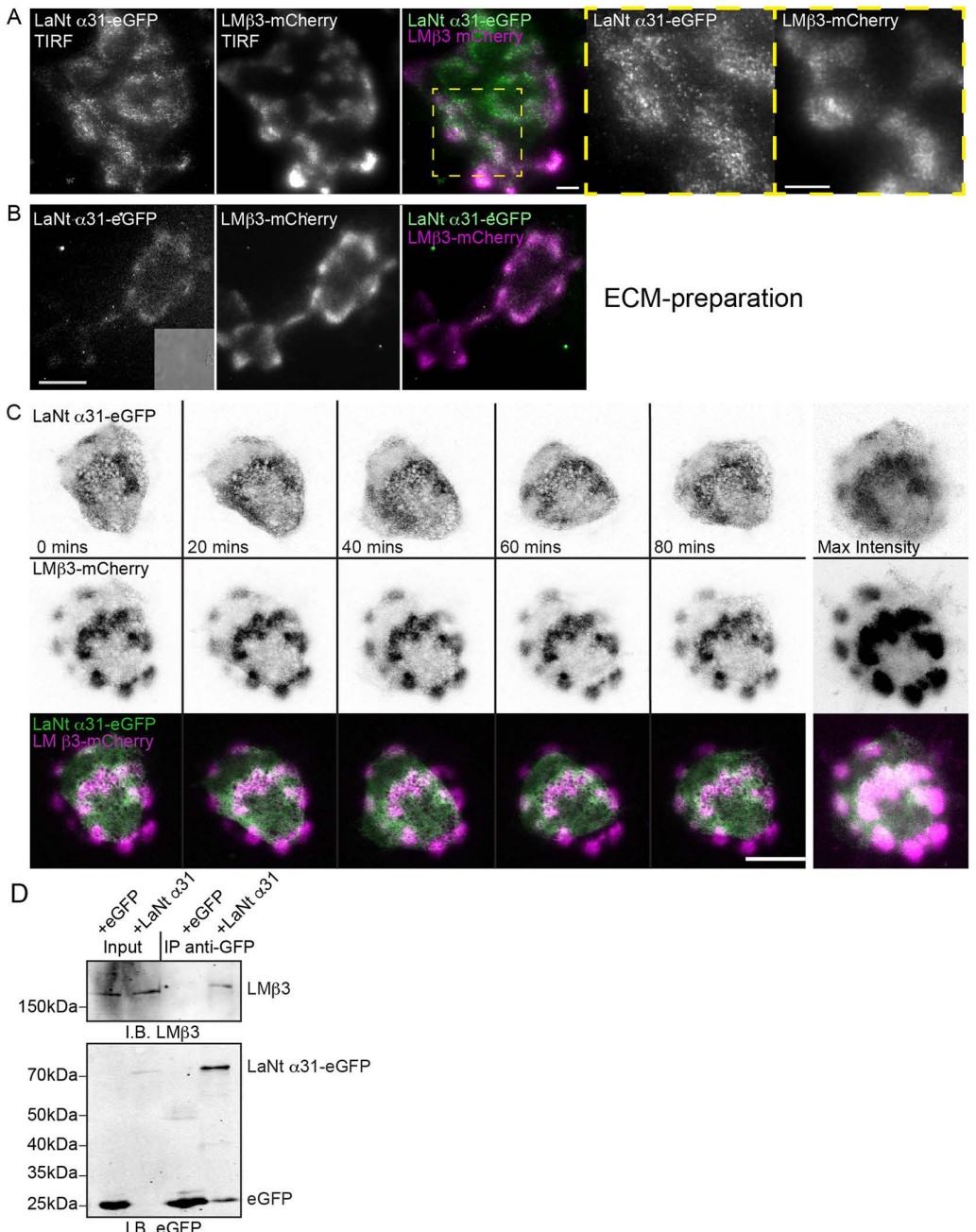

**Fig 2. LaNt α31 co-distributes and forms a complex with LMβ3. (A)** TIRF images of hTCEpi cells doubly transduced with LaNt α31-eGFP (left panel, pseudocoloured green in merge) and LMβ3-mCherry (second panel, pseudocoloured magenta in merge) imaged 16 h after plating on glass-bottomed dishes. Yellow boxed region is shown at higher magnification in panels to the right. **(B)** ECM preparation from doubly transduced cells generated by ammonium hydroxide mediated removal from cells 16 h after plating, phase contrast image in inset shows absence of cellular material. Area imaged equates matrix deposited by approximately 1-3 cells. (C) hTCEpi doubly transduced with LaNt α31-eGFP and LMβ3-mCherry were plated overnight on uncoated glass-bottomed dishes then imaged by confocal microscopy every 20 min over 16 **h. (C)** Representative images from 0 to 80 min of LaNt α31-eGFP (upper panels) and LMβ3-mCherry (middle panels), with signals inverted to aid visualisation. Bottom panel, merged images of LaNt a31-eGFP (pseudocoloured green) and LMβ3-mCherry (pseudocoloured magenta). Panels to far right are maximum intensity projection of the entire 16 h time course images. **(D)** Total protein was extracted from hTCEPi expressing either eGFP only (+eGFP) or LaNt α31-eGFP (+LaNt α31), and the eGFP-tag protein immunoprecipitated. Anti-GFP and anti-LMβ3 immunoblots of input, unbound fraction, wash fraction and immunoprecipitated fractions.

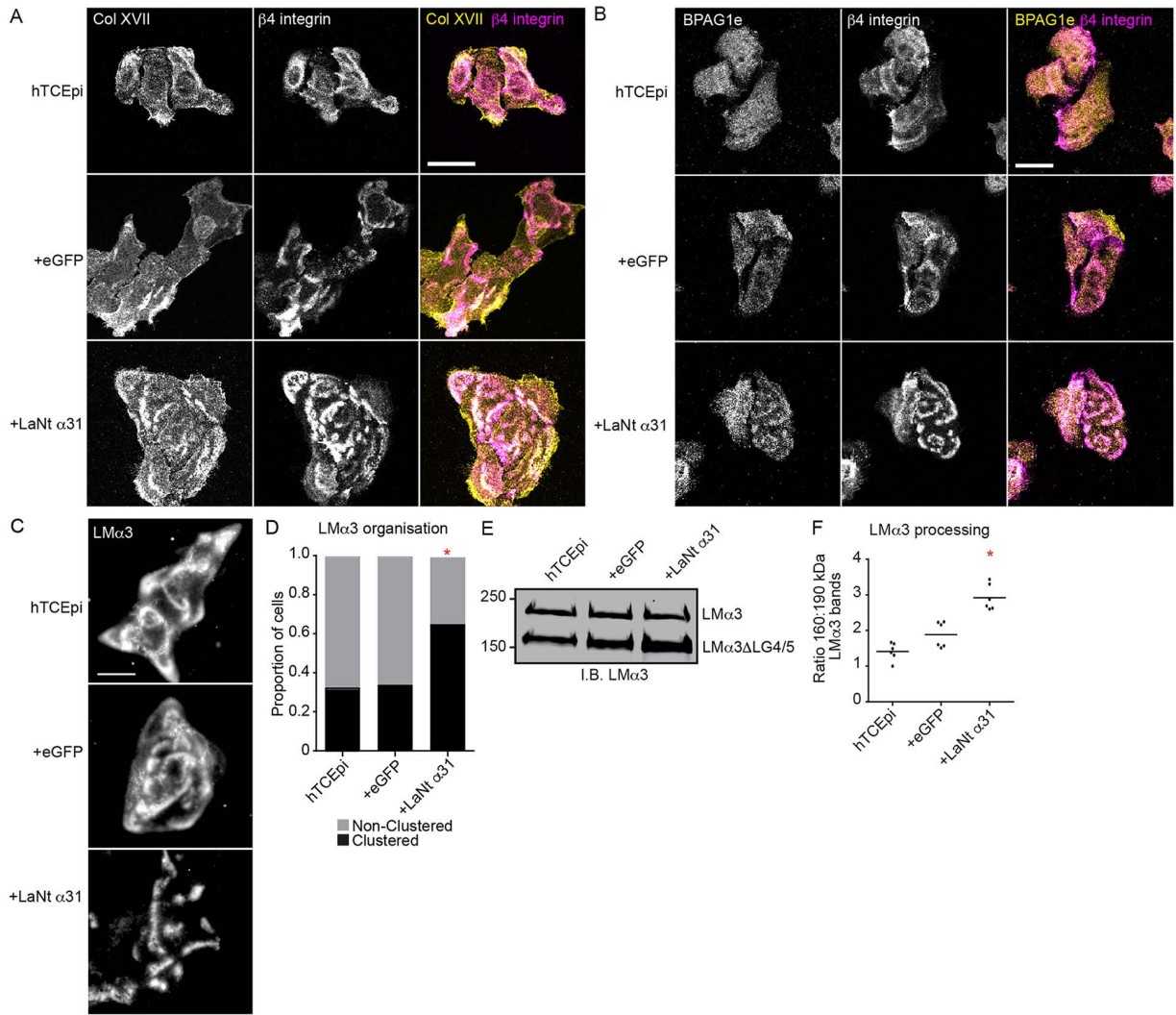

**Fig 3. LaNt α31 expression induces changes to LMα3 organisation and hemidesmosomes are more mature in LaNt α31 expressing corneal epithelial cells.** Non-transduced,+eGFP, or +LaNt α31 hTCEpi were plated on glass coverslips then fixed and processed for indirect immunofluorescence with antibodies against: **(A)** type XVII collagen (Col XVII) and b4 integrin; **(B)** bullous pemphigoid antigen 1e (BPAG1e) and b4 integrin; **(C)** LMα3. Scale bars 20 μm. **(D)** LMα3 staining was scored by independent observers blinded to the treatment conditions. Each cell image was scored as either "clustered" where there was clear separation between the discrete areas of staining, or "non-clustered" where the staining was predominantly in continuous arcs or swirls. Plotted is the proportion of the population from >100 cells in 3 independent experiments. With black bars representing the clustered population. * denote significant differences at p < 0.025 from untreated hTCEpi by 2-tailed binomial test with Bonferroni correction for multiple comparisons. **(E and F)** Non-transduced,+eGFP, or +LaNt α31 hTCEpi were plated overnight on tissue culture plastic and ECM extracts prepared by ammonium hydroxide-mediated hyperosmotic removal of cellular material. **(E)** Representative immunoblot with antibodies against LMα3. **(F)** Ratiometric quantification of 160 kDa band to 190 kDa band. Each point represents an independent experiment. * denotes p < 0.05 compared with both controls as determined by one-way ANOVA with Bonferroni post-hoc test.

## Materials and methods

### Experimental model

The limbal-derived corneal epithelial cell line hTCEpi cell line (S1 Fig) [27] was used throughout this study. These cells were derived from derived from a 62-year-old male's cornea. We used adenoviral constructs to express LaNt α31 with a C-terminal eGFP tag (+LaNt α31, Fig 1B) or eGFP-only (+eGFP).

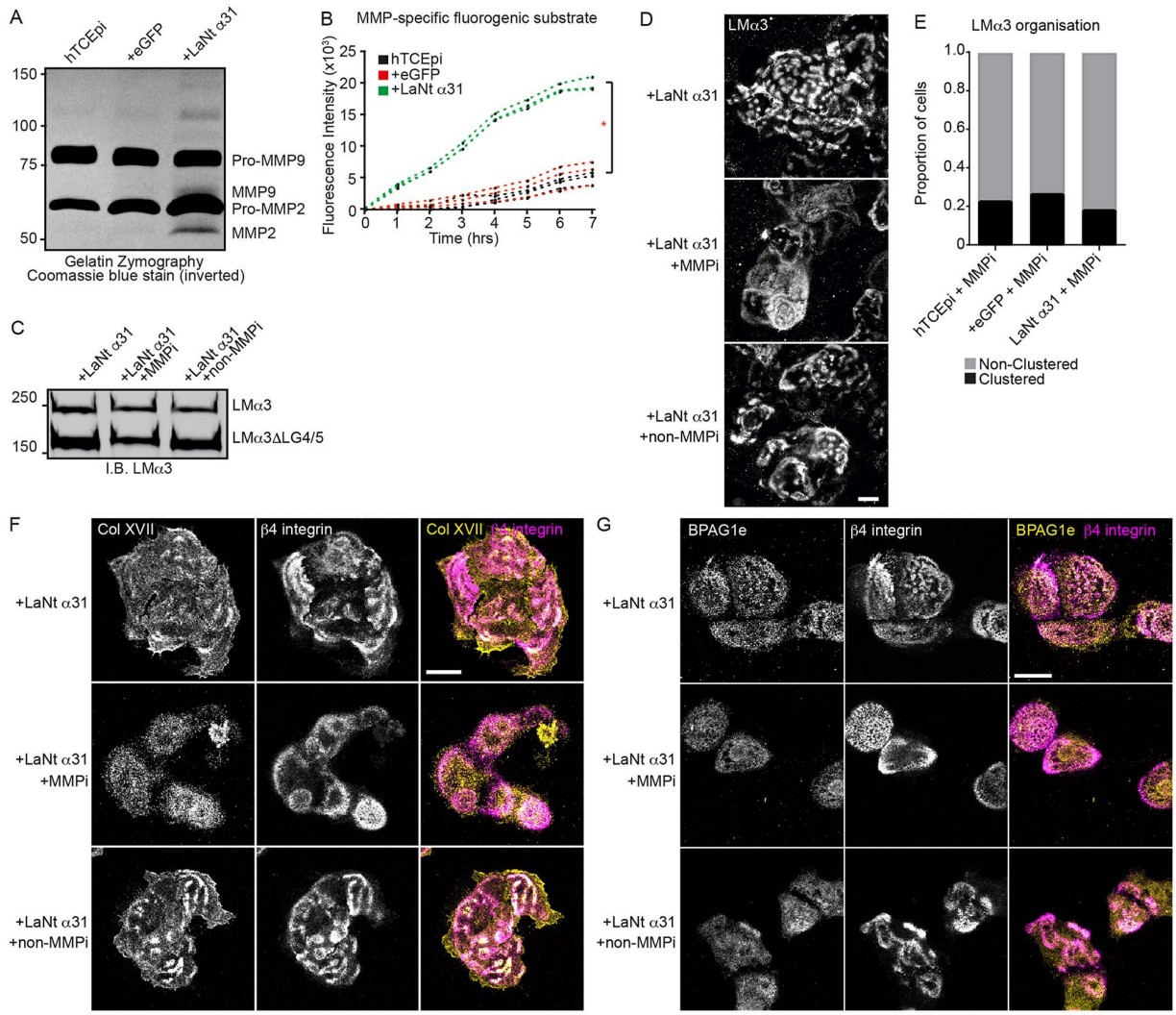

**Fig 4. Induced expression of LaNt α31 increases MMP activity, which is required for LMα3 clustering and hemidesmosome maturation effects.** **(A)** Colour-inverted gelatin zymography of conditioned media extracts generated from non-transduced, +eGFP, or +LaNt α31 hTCEpi. Indicated to the right of the gel image are the predicted sizes of pro and active forms of MMP2 and 9. **(B)** Conditioned media extracts were normalised for cell number after fixing and counting DAPI-stained nuclei, then incubated with the BML-P131-0001 peptide, which fluoresces only once cleaved by MMPs. The graph displays the fluorescence intensity measured over time with each line representing an independent experiment. * denotes p < 0.05 between +LaNt α31 and both controls, determined by ACOVA. **(C-F)** +LaNt α31 hTCEpi cells were plated overnight with broad spectrum MMP inhibitor (MMPi) or non-MMP protease inhibitor (non-MMPi) and extracellular matrixes prepared for western blotting **(C)** or coverslips processed for indirect immunofluorescence microscopy with antibodies against LMα3 **(D)** or type XVII collagen and β4 integrin **(F)**. Scale bar 20 µm. **(E)** quantification of LMα3 cluster vs non-cluster area of MMPi treated cells. Scoring performed by blinded scorers from 3 independent experiment, > 100 cells per treatment condition. Differences between groups did not reach statistical significance.

## Antibodies and other reagents

Rabbit monoclonal antibodies raised against paxillin (clone Y113, OriGene Cat# TA300443, RRID:AB_2174728), were used at 10 µg/ml for immunofluorescence microscopy (IF), mouse monoclonal antibodies raised against β4 integrin (clone M126, Abcam Cat# ab29042, RRID:AB_870635) were used at 4 µg/ml, and rabbit monoclonal antibodies raised against type XVII collagen (clone EPR18614, ab184996) were used at 10 µg/ml for IF and 0.25 µg/ml for immunoblotting (IB),

all from Abcam (Abcam, Cambridge, UK). Mouse monoclonal antibodies for IF against LMα3 (clone RG13) [41] used at 10 μg/ml and human monoclonal antibodies to BPAG1e (5E) [42] used at 1:2, were generous gifts from Jonathan Jones, Washington State University, WA. Mouse monoclonal antibodies raised against human LaNt α31 (clone 3E11) used at 1.9 μg/ml for IB were described previously. Mouse monoclonal antibodies raised against LMα3 (clone CL3112, HPA001895, RRID:AB_2637263) used at 0.5 μg/ml, LMα5 (clone 2F7, WH0003911M1, RRID:AB_1842250) used at 1 μg/ml, and against GFP (a mixture of clones 7.1 and 13.1, 11814460001, RRID:AB_390913) used at 0.08 μg/ml, were from Sigma-Aldrich (Sigma-Aldrich, St. Louis, Missouri, USA). Rabbit polyclonal antibodies raised against LMα1 (PA5–27271, RRID:AB_2544747) were used at 1 μg/ml, against LMα3 (PA5–29161, RRID:AB_2546637) at 0.46 μg/ml, against LMα2 (PA5–51110, RRID:AB_2636558) at 0.66 μg/ml, Alexa Fluor™ 647 Phalloidin (A22287, RRID:AB_2620155) used at 5 units/ml, were from ThermoFisher (ThermoFisher Scientific, Waltham, Massachusetts, USA). Alexa Fluor 594 nm, 647 nm or Cy5 conjugated goat anti-mouse and goat anti-rabbit secondary antibodies were obtained from Jackson Immunoresearch (Stratech, Ely, UK). Goat anti-mouse IRDye 800CW and goat anti-rabbit IRDye 680CW were obtained from LI-COR BioSciences (LI-COR BioSciences, Lincoln, Nebraska, USA).

For MMP inhibition, hydroxamate-based MMP inhibitor CT-1746 (N1-[2-(S)-(3,3-dimethylbutanamidyl)]-N4-hydroxy-2-(R)-[3-(4-chlorophenyl)-propyl]-succinamide) was diluted to 100 μM in culture media (UCB Celltech, Slough, U.K). For inhibition of serine, cysteine and aspartic proteases, P1860 inhibitor cocktail consisting of aprotinin, bestatin, E-64, leupeptin, pepstatin A was diluted to 1:1000 in culture media (Sigma-Aldrich).

For ECM coating for cell migration assays, human fibronectin from plasma and rat-tail collagen I (Sigma-Aldrich), and human LM332 and LM511 (Biolamina, UK) were used at 10 μg/ml, coating overnight at 4°C.

## Cell culture

Telomerase-immortalized human corneal epithelial cells, hTCEpi cells [27] (RRID: CVCL_AQ44), were cultured at 37°C with 5% $CO_2$ in keratinocyte-serum-free medium (KSFM) supplemented with bovine pituitary extract (BPE, 0.05 mg/ml), human recombinant Epidermal Growth Factor (5 ng/ml, ThermoFisher Scientific) and 0.15 mM $CaCl_2$ (Sigma-Aldrich).

## Adenovirus production and cell transduction

Adenoviral constructs inducing CMV-driven full-length human LaNt α31 with a C-terminal eGFP tag were described previously [6]. Adenoviruses inducing CMV driven expression of LMβ3-mCherry [43] and eGFP were kind gifts from Jonathan Jones, Washington State University, WA.

hTCEpi were cultured in T75 flasks (Greiner-Bio One, Stonehouse, Gloucestershire, UK) until 70% confluent or were seeded at $1 \times 10^6$ cells/plate in 100 mm plates for 24h. Cells were then transduced with eGFP, LaNt α31-eGFP or LMβ3-mCherry adenoviruses. Transduction efficiency was confirmed by fluorescence microscopy or flow cytometiry as being 60–90% at point of use. Experiments were conducted 24–48 h following transduction.

## SDS PAGE, immunoblotting and immunoprecipitation

For total protein extracts, $1 \times 10^6$ cells were seeded in 100 mm dishes (Greiner-BioOne) then lysed after 6 h or 24 h in urea/ sodium dodecyl sulfate (SDS) buffer (10 mM Tris-HCl pH 6.8, 6.7 M Urea, 1% SDS, 10% glycerol and bromophenol blue) containing 50 μM phenylmethylsulfonyl fluoride (PMSF) and 50 μM N-ethylmaleimide (all Sigma-Aldrich). For LMα3 ECM processing blots, cells were seeded at $2.5 \times 10^5$ cells/well in 6-well plates. After 16 h, cellular material was removed through exposure to 0.18% $NH_4OH$ for 5 min followed by extensive PBS washes [44], and ECM solubilised in urea/ SDS buffer. Before loading samples were sonicated (for total protein lysates), and 10% β-mercaptoethanol (final volume) added.

For immunoprecipitation 48 h after transduction cells were extracted in 0.1% SDS, 0.5% sodium deoxycholate, 1% Nonidet P-40, 150 mM NaCl, 1 mM $CaCl_2$ in 50 mM Tris-HCl, pH 7.5 with protease and phosphatase inhibitors (all

Sigma-Aldrich). Cell extracts were clarified by centrifugation, and a 50% slurry of Sepharose beads covalently conjugated with rabbit anti-GFP polyclonal antibodies (Abcam) added to the supernatant and incubated overnight at 4°C. Beads were washed in lysis buffer, collected by centrifugation and boiled in SDS-PAGE sample buffer (50 mM Tris-HCl pH 6.8, 10% glycerol, 1% SDS, 10% BME).

Proteins were separated by SDS-polyacrylamide gel electrophoresis (SDS-PAGE) using 7.5% or 10% polyacrylamide gels (1.5 M Tris, 0.4% w/v SDS, 7.5% 10% acrylamide/ bis-acrylamide; electrophoresis buffer: 25 mM Tris, 250 mM glycine, 0.1% w/v SDS, pH 8.5. All Sigma-Aldrich) and transferred to a nitrocellulose membrane using a Biorad TurboBlot system (BioRad, Hercules California, USA) or a BioRad wet transfer system (12.5 mM Tris, 125 mM glycine, 0.05% SDS, 20% ethanol), and blocked for 1 h at room temperature in Odyssey°TBS-Blocking Buffer (LI-COR BioSciences) or in 5% (w/ v) Marvel skimmed milk in TBS (Tesco, Hertfordshire, UK). The membranes were probed with primary antibodies diluted in blocking buffer overnight at 4°C, washed 3 x 5 min in TBS with 0.1% Tween (TBS-T) then probed for 1 h at room temperature with IRDye° 800CW or IRDye° 680CW conjugated secondary antibodies (LI-COR) or HRP- conjugated secondary antibodies diluted in blocking buffer (0.05 µg/ml). Membranes were then washed for 3 x 5 min in TBS-T and imaged using an Odyssey° CLx 9120 Infrared Imaging System (LI-COR), or imaged on a BioRad Chemidoc after incubating with Immobilon chemiluminescence substrate.

## Gap closure, cell migration assays, and cell morphology analyses

For gap closure assays, cells were seeded into ibidi® 2-well culture inserts (Ibidi, Martinsried, Germany); at $7.0 \times 10^4$ cells/ well. Culture inserts were carefully removed after 6 h, cell debris washed away, and the gap margin imaged using bright-field optics on a Nikon TiE epifluorescence microscope with a 10X objective (Nikon, Tokyo, Japan). Gap closure was measured as a percentage relative to starting area using the freehand tool in Image J (NIH, Bethesda, Massachusetts, USA).

For cell morphology analyses and low-density migration assays, cells were seeded at $2.5 \times 10^4$ cells/well onto uncoated 24-well plates (Greiner-Bio One).

For morphology analyses, phase-contrast images and formaldehyde-fixed, phalloidin-stained images were acquired on a Nikon TiE microscope and analyzed using ImageJ software. The phase-contrast images were used for analyses. Cell perimeters were manually traced to define cell area; cell length was measured as the longest linear axis, with width measured at widest point at right angles to the length measurement. The aspect ratio was defined as the ratio of width to length. For low-density migration assays, cells were imaged every 2 min over a 2 h period, using a 20X objective on a Nikon TiE fluorescent microscope, then individual cells tracked using the MTrackJ plugin on ImageJ. Speed (total distance travelled/time) and processivity (total distance/ distance from the origin) were calculated for each cell.

## Immunofluorescence microscopy

$7.5 \times 10^4$ (low density) or $1.5 \times 10^5$ (high density) cells were seeded onto no. #1 round 16 mm glass coverslips (Pyramid innovations Ltd, Polegate, UK) in 12-well plates (Greiner-BioOne) and cultured overnight for approximately 16 h before fixing and extracting in ice-cold methanol for 15 min, or in 3.7% formaldehyde (Sigma-Aldrich) for 10 min followed by 10 min in 0.2% Triton X-100 in PBS to permeabilise (Sigma-Aldrich). For ECM-only analysis, cellular material was removed with 0.18% $NH_4OH$ for 5 min followed by extensive PBS washes prior to fixation. Primary antibodies were diluted in PBS with 10% normal goat serum (Jackson Labs) and incubated at 4°C overnight; coverslips were then washed 3x for 5 min with PBS containing 0.05% tween-20 (PBS-T, Sigma-Aldrich) before probing for 1 h at room temperature with Alexa Fluor 594nm or 647nm/ Cy5 conjugated secondary antibodies diluted in PBS. Coverslips were washed 3x for 5 min with PBS-T, counterstained with DAPI (ThermoFisher Scientific) for 10 min, rinsed thoroughly with deionised $H_2O$ and mounted with Vectashield (Vector Laboratories, Burlingame, California, USA). Images were obtained using a Zeiss LSM800 confocal microscope (Zeiss, Cambridge, UK).

LMα3 staining area was measured for 50 regions of staining per cell treatment using the freehand selection tool on ImageJ. For type XVII collagen colocalisation with integrin β4, Manders' correlation coefficients were calculated using the Coloc 2 plugin on ImageJ for all pixels above channel thresholds.

### Live protein imaging

For TIRF microscopy and new matrix deposition assays, $1 \times 10^5$ hTCEpi cells, doubly transduced with LaNt α31 GFP and LMβ3 mCherry adenoviruses were seeded onto uncoated 35 mm glass-bottomed dishes (MatTek Corporation, Ashland, Massachusetts, USA) then imaged after 16 h using a Zeiss 510 Multiphoton 2 confocal microscope or Zeiss LSM880 confocal laser scanning microscope with TIRF capability. In live experiments, images were acquired using a 63X objective every 20 min over 16 h. Images were processed using ImageJ.

### MMP activity assays

hTCEpi cells were seeded at $8 \times 10^5$ cells per dish in 60 mm dishes (Greiner-BioOne) and cultured in 2 mL BPE-free KSFM media for 48 h. Conditioned media was collected and concentrated to 1 mL using Amicon Ultra-0.5 centrifuge filters with 10 kDa pore size (Sigma-Aldrich). Cell counts were achieved by fixing the cells in 3.7% formaldehyde for 10 min, staining with 0.25 µg/ml DAPI for 10 min (ThermoFisher Scientific), imaging at the centre of the dish using 4x objective on a Nikon TiE microscope, then counting using the analyze particles tool on image J. The final volumes of harvested media normalized for cell number, and supplemented with 0.1% bovine serum albumin (BSA, Sigma-Aldrich) to stabilise MMPs.

For gelatin zymography assays, concentrated normalized conditioned media was mixed with SDS-PAGE sample buffer without BME and ran on a 1 mg/ml gelatin-containing 10% polyacrylamide gel (gelatine, Sigma-Aldrich). The gels were washed for 2 h in renaturing buffer (50 mM Tris-HCL containing 2.5% Triton-x 100) and incubated at 37°C in zymography buffer (50 mM Tris-HCL containing 5 mM $CaCl_2$) for 24 h. The gels were then stained overnight at 4°C, destained for 2 h at room temp (Coomassie Brilliant Blue R-250 staining kit, BIO-RAD), and imaged on a ChemiDoc MP (BIO-RAD).

For fluorogenic substrate assays, conditioned media was diluted 1:1 zymography buffer containing 10 µM peptide (BML-P131-0001, Dnp-Pro-β-cyclohexyl-Ala-Gly-Cys(Me)-His-Ala-Lys(Nma)-$NH_2$, Enzo Lifesciences, Exeter, UK) and 10 mM $CaCl_2$ (final conc. 5 µM/ 5 mM, respectively) and 100 µl added in triplicate to a black 96-well plates (Corning, Somerville Massachusetts, USA). Plates were read every hour for 7 h at an excitation/ emission spectra of 280 nm/ 350 nm on a SPECTROstar-nano plate reader (BMG LABTECH Ltd, Bucks, UK).

### Statistical analyses

Quantification of experiments was derived from a minimum of 3 independent experiment repeats. One-way ANOVA and Tukey's HSD or Bonferroni post hoc analysis or linear regression and ANCOVA were used to analyze and interpret the data using GraphPad Prism 6 (GraphPad Software, San Diego, California, USA).

### Supporting information

**S1 Fig. Cell line validation.** hTCEpi corneal epithelial cells, human retinal pericytes (HRP) or human dermal fibroblasts were plated overnight on glass coverslips then fixed and processed with antibodies against keratin 12 and vimentin. Scale bar 20 µm.
(TIF)

**S2 Fig. Corneal epithelial cells induced to express LaNt α31 display decreased cell migration.** hTCEpi cells were plated at confluence ibidi® 2-well culture inserts, removing after 6 h. (A) Representative images from immediately after removing (T0 upper panels) and after 16 h of recovery (T16 lower panels), yellow lines indicate gap margins. Scale bar 100µm. (B) Gap area closure measured 16 h after removing inerts plotted as percentage of the initial gap area with each

point representing an independent experiment with either 3 technical repeats per experiment. * denotes $p < 0.05$ compared with controls determined by one-way ANOVA followed by Bonferroni post hoc test.
(TIF)

**S3 Fig. Laminin isoform transcript expression comparison.** Total RNA was extracted from cells and RT-qPCR performed with primers specific to each laminin encoding gene. Bar chart represents the proportion of each isoform expressed, separated into LAMA, LAMB, and LAMC laminin isoforms.
(TIF)

**S4 Fig. Focal adhesions are mislocalised in LaNt α31 expressing corneal epithelial cells.** (A) Non-transduced, +eGFP, or +LaNt α31 hTCEpi were plated on glass coverslips then fixed and processed for indirect immunofluorescence with antibodies against paxillin and with phalloidin to label filamentous actin (F-actin). Scale bars 20 μm. (B) and (D) total protein extracts from the indicated cell treatments were immunoblotted with antibodies against paxillin and tyrosine 118 phosphorylated paxillin (Y118), or anti-focal adhesion kinase (FAK) or phosphorylated tyrosine 397 FAK (Y397). Dot plots in (C) and (E) are derived from densitometry analyses of immunoblots of the intensity of phosphorylated band/total protein. Each point represents an independent experiment, with lines indicating mean. Differences between groups did not reach statistical significance.
(TIF)

**S5 Fig. Uncropped blots and gel images.**
(TIF)

**S1 File. LaNt α31 rapidly forms and disassembles clusters at LMβ3-mCherry deposition sites.** hTCEpi co-transduced with LaNt α31-eGFP and LMβ3-mCherry were plated overnight on uncoated glass-bottomed dishes and imaged by confocal microscopy every 20 min. Left panel LaNt α31-eGFP, middle panel LMβ3-mCherry with signals inverted, right panel, merged LaNt α31-eGFP (pseudocoloured green) and LMβ3 mCherry (pseudocoloured magenta).
(AVI)

**S2 File. Inclusivity-in-global-research-questionnaire.**
(DOCX)

**S3 File. LaNt PLOSOne 2025 numerical data.**
(XLSX)

## Acknowledgments

The authors would like to acknowledge Jonathan Jones and Susan Hopkinson (Washington State University) and Robert Lavker (Northwestern University) for generous gifts of reagents and cells. Technical help was greatly appreciated from David Mason, Marco Marcello, Violaine See and Joanna Majchrowska at the Centre for Cell Imaging at the University of Liverpool.

## Author contributions

**Conceptualization:** Lee D. Troughton, Valentina Iorio, Kevin J. Hamill.

**Data curation:** Lee D. Troughton, Valentina Iorio, Liam Shaw, Djamilla Simoens, Conor J. Sugden, Natasha D. Chavda, Peter Wilson, Stefano Sala, Kevin J. Hamill.

**Formal analysis:** Lee D. Troughton, Valentina Iorio, Liam Shaw, Djamilla Simoens, Conor J. Sugden, Natasha D. Chavda, Andrea E. Conway, Peter Wilson, Kevin J. Hamill.

**Funding acquisition:** Lee D. Troughton, Simon Kaja, Kazuhiro Yamamoto, Kevin J. Hamill.

**Investigation:** Lee D. Troughton, Valentina Iorio, Stefano Sala, Kevin J. Hamill.

**Methodology:** Lee D. Troughton, Valentina Iorio, Kazuhiro Yamamoto, Kevin J. Hamill.

**Project administration:** Lee D. Troughton, Valentina Iorio, Kevin J. Hamill.

**Resources:** Simon Kaja, Kazuhiro Yamamoto, Kevin J. Hamill.

**Supervision:** Kevin J. Hamill.

**Writing – original draft:** Lee D. Troughton, Valentina Iorio, Kevin J. Hamill.

**Writing – review & editing:** Lee D. Troughton, Liam Shaw, Djamilla Simoens, Conor J. Sugden, Natasha D. Chavda, Andrea E. Conway, Peter Wilson, Stefano Sala, Simon Kaja, Kazuhiro Yamamoto, Kevin J. Hamill.

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
