## [Decision Letter · Decision Letter 0]

22 Oct 2025

Dear Dr. Hamill,

Thank you for submitting your manuscript to PLOS ONE. After careful consideration, we feel that it has merit but does not fully meet PLOS ONE’s publication criteria as it currently stands. Therefore, we invite you to submit a revised version of the manuscript that addresses the points raised during the review process.

We look forward to receiving your revised manuscript.

Kind regards,

Alexander V Ljubimov, Ph.D.

Academic Editor

PLOS ONE

Journal Requirements:

https://journals.plos.org/plosone/s/file?id=ba62/PLOSOne_formatting_sample_title_authors_affiliations.pdf
**.**

“This work was supported by funding from the Biotechnology and Biological Sciences Research Council (K.H BB/L020513/1 and BB/P025773/1, and a BBSRC DTP PhD studentship), Northwest Cancer Research Fund (K.H), Fight For Sight (K.H New lecturers' award), British Skin Foundation (K.H PhD studentship), Foundation for the Prevention of Blindness (K.H), ECMage impact award (K.H), Versus Arthritis Career Development Fellow (K.Y 21447), Versus Arthritis Bridging Fellow (K.Y 23137). Additional support from the National Institute of Health (S.K grant R24/EY032440), the Dr. John P. and Therese E. Mulcahy Endowed Professorship in Ophthalmology (S.K), the Richard A. Perritt M.D. Charitable Foundation and the Illinois Society for the Prevention of Blindness are gratefully acknowledged (L.T). The funders played no role in the study design, data collection and analysis, decision to publish, or preparation of this manuscript.

https://www.ukri.org/councils/bbsrc/

https://www.fightforsight.org.uk/

https://www.britishskinfoundation.org.uk/

https://nwcr.org/

https://versusarthritis.org/

https://www.nih.gov/

https://eyehealthillinois.org/”

“This work was supported by funding from the Biotechnology and Biological Sciences Research Council (BB/L020513/1 and BB/P025773/1, and a BBSRC DTP PhD studentship), Northwest Cancer Research Fund, Fight For Sight (New lecturers' award), British Skin Foundation (PhD studentship), Foundation for the Prevention of Blindness, ECMage impact award, Versus Arthritis Career Development Fellow (21447), Versus Arthritis Bridging Fellow (23137). Additional support from the National Institute of Health (grant R24/EY032440), the Dr. John P. and Therese E. Mulcahy Endowed Professorship in Ophthalmology, the Richard A. Perritt M.D. Charitable Foundation and the Illinois Society for the Prevention of Blindness are gratefully acknowledged.”

“This work was supported by funding from the Biotechnology and Biological Sciences Research Council (K.H BB/L020513/1 and BB/P025773/1, and a BBSRC DTP PhD studentship), Northwest Cancer Research Fund (K.H), Fight For Sight (K.H New lecturers' award), British Skin Foundation (K.H PhD studentship), Foundation for the Prevention of Blindness (K.H), ECMage impact award (K.H), Versus Arthritis Career Development Fellow (K.Y 21447), Versus Arthritis Bridging Fellow (K.Y 23137). Additional support from the National Institute of Health (S.K grant R24/EY032440), the Dr. John P. and Therese E. Mulcahy Endowed Professorship in Ophthalmology (S.K), the Richard A. Perritt M.D. Charitable Foundation and the Illinois Society for the Prevention of Blindness are gratefully acknowledged (L.T). The funders played no role in the study design, data collection and analysis, decision to publish, or preparation of this manuscript.

https://www.ukri.org/councils/bbsrc/

https://www.fightforsight.org.uk/

https://www.britishskinfoundation.org.uk/

https://nwcr.org/

https://versusarthritis.org/

https://www.nih.gov/

https://eyehealthillinois.org/”

Additional Editor Comments (if provided):

The paper has been reviewed and some minor concerns voiced. Please revise the paper according to the critique.

1. Please revisit Figure 4F-G (lines 234-236) and add missing Fig.4G.

2. “For gelatin zymography assays, concentrated normalized conditioned media…” Please specify how it was normalized.

3. It is customary to call skin epithelial cells keratinocytes. However, corneal counterparts are called corneal epithelial cells.

4. Adenovirus has been shown to be toxic for cultured primary human limbal epithelial cells. Please confirm that no toxicity at the used doses was observed (like with Apopnexin assay).

5. Whereas MMP-9 was mostly present in the pro-form, MMP-2 showed clear activation after LaNt�31 addition. Please discuss.

6. Please explain in more detail how the zymograms were obtained. Typically, the gel is stained with Coomassie, then incubated with the extracts/media and the enzyme bands are seen as white against the blue background. Here, the bands are black against white background. Was the picture color-reversed or another method used?

7. Supplemental figure 1. It is unusual to see K12 expression on hTERT-immortalized HCEC. Please comment that they could be more differentiated as compared to primary limbal cells.

8. For readers not familiar with HD structure, please mention that the used proteins, Type XVII collagen and BPAG, are HD structural components and markers.

Reviewers' comments:

Reviewer's Responses to Questions

**Comments to the Author**

1. Is the manuscript technically sound, and do the data support the conclusions?

Reviewer #1: Yes

2. Has the statistical analysis been performed appropriately and rigorously?

Reviewer #1: Yes

3. Have the authors made all data underlying the findings in their manuscript fully available?

Reviewer #1: Yes

4. Is the manuscript presented in an intelligible fashion and written in standard English?

Reviewer #1: Yes

Reviewer #1: The data presented characterize LaNt α31 as a regulator of epithelial homeostasis via modulation of cell–matrix adhesion and LM332 architecture and suggest that LaNt α31 could serve as a molecular switch from migratory to adhesive epithelial conditions.

The results obtained were presented in clear and well-documented manner, with the exception of the paragraph describing increased co-localisation of β4 integrin with type XVII collagen and BPAG1e rescued by MMP inhibition but not by non-MMP inhibitors (Figure 4F-G) (lines 234-236), as there was no Fig.4G presented in the manuscript.

**Do you want your identity to be public for this peer review?** For information about this choice, including consent withdrawal, please see our Privacy Policy

Reviewer #1: No

---

## [Author Response · Author response to Decision Letter 1]

21 Nov 2025

We appreciate all the reviewer’s comments and have addressed each point for your consideration, below.

1. Please revisit Figure 4F-G (lines 234-236) and add missing Fig.4G.

We have corrected this issue, adding panel 4G to figure 4.

2. “For gelatin zymography assays, concentrated normalized conditioned media…” Please specify how it was normalized.

After collecting condition media, the cells were fixed and DAPI-stained nuclei counted. Media volume was then adjusted based on that count. We have described this in the method section and have also added it to the figure legend for figure 4.

3. It is customary to call skin epithelial cells keratinocytes. However, corneal counterparts are called corneal epithelial cells.

We agree and have changed keratinoyctes to corneal epithelial cells throughout.

4. Adenovirus has been shown to be toxic for cultured primary human limbal epithelial cells. Please confirm that no toxicity at the used doses was observed (like with Apopnexin assay).

We appreciate this suggestion. hTCEpi are a limbal-derived cell line, immortalised, more mature epithelial cells. We have not experienced any evidence of toxicity using adenovirus with this line. Throughout, we have used a GFP-only adenovirus as a viral transduction and protein expression control. The GFP-virus conditions displayed no significant difference to the non-transduced hTCEpi in any experiments.

5. Whereas MMP-9 was mostly present in the pro-form, MMP-2 showed clear activation after LaNta31 addition. Please discuss.

We have included supplemental figure 5, which provides all our raw blots and gelatin zymography images. You can perhaps better appreciate here that we see an increase in activity for MMP2 and MMP9. The currently literature is not definitive on whether laminin LG3/4 linker is cleaved by MMP2 or MMP9, although both appear to be capable, therefore we are deliberately not making a statement along those lines. We have, however, stated in the discussion that future should focus on investigating the relationship between LaNt α31’s interaction with LM332 and MMPs.

6. Please explain in more detail how the zymograms were obtained. Typically, the gel is stained with Coomassie, then incubated with the extracts/media and the enzyme bands are seen as white against the blue background. Here, the bands are black against white background. Was the picture color-reversed or another method used?

You are correct, this is an inverted image as we believe the contrast allows for better visualisation. We have now clarified this in the figure legend and indicated in figure 4 panel A (and supplemental figure 5) itself to highlight the inversion. The zymography methods are described in the method section.

7. Supplemental figure 1. It is unusual to see K12 expression on hTERT-immortalized HCEC. Please comment that they could be more differentiated as compared to primary limbal cells.

To our knowledge, these cells express both keratin 3/12 and 5/14, although it is true that they likely have higher expression of 5/14 as they are not stratified. We deliberately hadn’t commented on the differentiation state of these cells as we haven’t looked in detail at their keratin expression profile. We included this figure as supplement to demonstrate they are keratin positive epithelial cells. We have however added a line in 112/113 stating they appear to be a more mature corneal epithelial line.

8. For readers not familiar with HD structure, please mention that the used proteins, Type XVII collagen and BPAG, are HD structural components and markers.

Thank you for pointing this out. With have included this in the results section, line 188.

Additionally, we have removed the funding body information from the acknowledgements and amended the funding disclosure statement as requested, uploading separately as “Funding statement”.

We have also compiled all the numerical data used in a single excel file which will be made available. All the uncropped blots and gel images have been included in the manuscript in a new figure (Supplemental Figure 5).

---

## [Editor Report · Decision Letter 1]

27 Nov 2025

Laminin N-terminus α31 regulates corneal epithelial cell adhesion and migration through modifying the organization and proteolytic processing of laminin 332.

PONE-D-25-45306R1

Dear Dr. Hamill,

We’re pleased to inform you that your manuscript has been judged scientifically suitable for publication and will be formally accepted for publication once it meets all outstanding technical requirements.

Kind regards,

Alexander V Ljubimov, Ph.D.

Academic Editor

PLOS ONE
---

## [Editor Report · Acceptance letter]

PONE-D-25-45306R1

PLOS One

Dear Dr. Hamill,

I'm pleased to inform you that your manuscript has been deemed suitable for publication in PLOS One. Congratulations! Your manuscript is now being handed over to our production team.

Kind regards,

on behalf of

Dr. Alexander V Ljubimov

Academic Editor

PLOS One